# Block Coordinate Plug-and-Play Methods for Blind Inverse Problems

**Weijie Gan**
Washington University in St. Louis
`weijie.gan@wustl.edu`

**Shirin Shoushtari**
Washington University in St. Louis
`s.shirin@wustl.edu`

**Yuyang Hu**
Washington University in St. Louis
`h.yuyang@wustl.edu`

**Jiaming Liu**
Washington University in St. Louis
`jiaming.liu@wustl.edu`

**Hongyu An**
Washington University in St. Louis
`hongyuan@wustl.edu`

**Ulugbek S. Kamilov**
Washington University in St. Louis
`kamilov@wustl.edu`

## Abstract

Plug-and-play (PnP) prior is a well-known class of methods for solving imaging inverse problems by computing fixed-points of operators combining physical measurement models and learned image denoisers. While PnP methods have been extensively used for image recovery with known measurement operators, there is little work on PnP for solving blind inverse problems. We address this gap by presenting a new block-coordinate PnP (BC-PnP) method that efficiently solves this joint estimation problem by introducing learned denoisers as priors on both the unknown image and the unknown measurement operator. We present a new convergence theory for BC-PnP compatible with blind inverse problems by considering *nonconvex* data-fidelity terms and *expansive* denoisers. Our theory analyzes the convergence of BC-PnP to a stationary point of an *implicit* function associated with an *approximate* minimum mean-squared error (MMSE) denoiser. We numerically validate our method on two blind inverse problems: automatic coil sensitivity estimation in magnetic resonance imaging (MRI) and blind image deblurring. Our results show that BC-PnP provides an efficient and principled framework for using denoisers as PnP priors for jointly estimating measurement operators and images.

## 1 Introduction

Many problems in computational imaging, biomedical imaging, and computer vision can be formulated as *inverse problems* involving the recovery of high-quality images from low-quality observations. Imaging inverse problems are generally ill-posed, which means that multiple plausible clean images could lead to the same observation. It is thus common to introduce prior models on the desired images. While the literature on prior modeling of images is vast, current methods are often based on *deep learning (DL)*, where a deep model is trained to map observations to images [1–3].

*Plug-and-play (PnP) priors* [4, 5] is one of the most widely-used DL frameworks for solving imaging inverse problems. PnP methods circumvent the need to explicitly describe the full probability density of images by specifying image priors using image denoisers. The integration of state-of-the-art deep denoisers with physical measurement models within PnP has been shown to be effective in a number of inverse problems, including image super-resolution, phase retrieval, microscopy, and

37th Conference on Neural Information Processing Systems (NeurIPS 2023).

medical imaging [6–13] (see also recent reviews [14, 15]). Practical success of PnP has also motivated novel extensions, theoretical analyses, statistical interpretations, as well as connections to related approaches such as score matching and diffusion models [16–29].

Despite the rich literature on PnP, the existing work on the topic has primarily focused on the problem of image recovery where the measurement operator is known exactly. There is little work on PnP for *blind* inverse problems, where both the image and the measurement operator are unknown. This form of blind inverse problems are ubiquitous in computational imaging with well-known applications such as blind deblurring [30] and parallel magnetic resonance imaging (MRI) [31]. In this paper, we address this gap by developing a new PnP approach that uses denoisers as priors over both the unknown measurement model and the unknown image, and efficiently solves the joint estimation task as a *block-coordinate PnP (BC-PnP)* method. While a variant of BC-PnP was proposed in the recent paper [21], it was never used for jointly estimating the images and the measurement operators. Additionally, the convergence theory in [21] is inadequate for blind inverse problems since it assumes convex data-fidelity terms and nonexpansive denoisers. We present a new convergence analysis applicable to *nonconvex* data-fidelity terms and *expansive* denoisers. Our theoretical analysis provides explicit error bounds on the convergence of BC-PnP for *approximate* minimum mean squared error (MMSE) denoisers under a set of clearly specified assumptions. We show the practical relevance of BC-PnP by solving joint estimation problems in blind deblurring and accelerated parallel MRI. Our numerical results show the potential of denoisers to act as PnP priors over the measurement operators as well as images. Our work thus addresses a gap in the current PnP literature by providing a new efficient and principled framework applicable to a wide variety of blind imaging inverse problems.

All proofs and some details that have been omitted for space appear in the supplementary material.

## 2  Background

**Inverse Problems.** Many imaging problems can be formulated as inverse problems where the goal is to estimate an unknown image $\boldsymbol{x} \in \mathbb{R}^n$ from its degraded observation $\boldsymbol{y} = \boldsymbol{A}\boldsymbol{x} + \boldsymbol{e}$, where $\boldsymbol{A} \in \mathbb{R}^{m \times n}$ is a measurement operator and $\boldsymbol{e} \in \mathbb{R}^m$ is the noise. A common approach for solving inverse problems is based on formulating an optimization problem

$$\widehat{\boldsymbol{x}} \in \arg\min_{\boldsymbol{x} \in \mathbb{R}^n} f(\boldsymbol{x}) \quad \text{with} \quad f(\boldsymbol{x}) = g(\boldsymbol{x}) + h(\boldsymbol{x}) \,, \tag{1}$$

where $g$ is the data-fidelity term that quantifies consistency with the observation $\boldsymbol{y}$ and $h$ is the regularizer that infuses a prior on $\boldsymbol{x}$. For example, a widely-used data-fidelity term and regularizer in computational imaging are the least-squares $g(\boldsymbol{x}) = \frac{1}{2} \|\boldsymbol{A}\boldsymbol{x} - \boldsymbol{y}\|_2^2$ and the total variation (TV) functions $h(\boldsymbol{x}) = \tau \|\boldsymbol{D}\boldsymbol{x}\|_1$, where $\boldsymbol{D}$ is the image gradient, and $\tau > 0$ a regularization parameter.

The traditional inverse problem formulations assume that the measurement operator $\boldsymbol{A}$ is known exactly. However, in many applications, it is more practical to model the measurement operator as $\boldsymbol{A}(\boldsymbol{\theta})$, where $\boldsymbol{\theta} \in \mathbb{R}^p$ are unknown parameters to be estimated jointly with $\boldsymbol{x}$. This form of inverse problems are often referred to as *blind* inverse problems and arise in a wide-variety of applications, including pralellel MRI [32–37], blind deblurring [30, 38, 39], and computed tomography [40–43].

**DL.** There is a growing interest in DL for solving imaging inverse problems [1–3]. Instead of explicitly defining a regularizer, DL approaches for solving inverse problems learn a mapping from the measurements to the desired image by training a convolutional neural network (CNN) to perform regularized inversion [44–48]. Model-based DL (MBDL) has emerged as powerful DL framework for inverse problems that combines the knowledge of the measurement operator with an image prior specified by a CNN (see reviews [3, 49]). The literature of MBDL is vast, but some well-known examples include PnP, regularization by denoising (RED), deep unfolding (DU), compressed sensing using generative models (CSGM), and deep equilibrium models (DEQ) [50–54]. All these approaches come with different trade-offs in terms of imaging performance, computational and memory complexity, flexibility, need for supervision, and theoretical understanding.

The literature on DL approaches for blind inverse problems is broad, with many specialized methods developed for different applications. While an in-depth review would be impractical for this paper, we mention several representative approaches adopted in prior work. The direct application of DL to predict the measurement operator from the observation was explored in [55, 56]. Deep image prior (DIP) was used as a learning-free prior to regularize the image and the measurement operator

in [57, 58]. Generative models, including both GANs and diffusion models, have been explored as regularizers for blind inverse problems in [59, 60]. Other work considered the use of a dedicated neural network to predict the parameters of the measurement operator, adoption of model adaptation strategies, and development of autocalibration methods based on optimization [34–37, 61–64].

**PnP.** PnP [4, 5] is one of the most popular MBDL approaches based on using deep denoisers as imaging priors (see also recent reviews [14, 15]). For example, the proximal gradient method variant of PnP (referred to as PnP-ISTA in this paper) can be formulated as a fixed-point iteration [65]

$$\boldsymbol{x}^k \leftarrow \mathsf{D}_\sigma\left(\boldsymbol{z}^k\right) \quad \text{with} \quad \boldsymbol{z}^k \leftarrow \boldsymbol{x}^{k-1} - \gamma\nabla g(\boldsymbol{x}^{k-1}), \tag{2}$$

where $\mathsf{D}_\sigma$ is a denoiser with a parameter $\sigma > 0$ for controlling its strength and $\gamma > 0$ is a step-size. The theoretical convergence of PnP-ISTA has been explored for convex functions $g$ using monotone operator theory [20, 22] as well as for nonconvex functions based on interpreting the denoiser as a MMSE estimator [23]. The analysis in this paper builds on the convergence theory in [23] that uses an elegant formulation by Gribonval [66] establishing a direct link between MMSE estimation and regularized inversion. Many variants of PnP have been developed over the past few years [6–12], which has motivated an extensive research on its theoretical properties [16,18,20,22,23,27–29,67–70].

*Block coordinate regularization by denoising (BC-RED)* is a recent PnP variant for solving large-scale inverse problems by updating along a subset of coordinates at every iteration [21]. BC-RED is based on regularization by denoising (RED), another well-known variant of PnP that seeks to formulate an explicit regularizer for a given image denoiser [17, 19]. BC-RED was applied to several non-blind inverse problems and was theoretically analyzed for convex data-fidelity terms.

PnP was extended to blind deblurring in [71, 72] by considering an additional prior on blur kernels that promotes sparse and nonnegative solutions. PnP was also applied to holography with unknown phase errors by using the Gaussian Markov random field model as the prior for the phase errors [73]. *Calibrated RED (Cal-RED)* [43] is a recent related extension of RED that calibrates the measurement operator during RED reconstruction by combining the traditional RED updates over an unknown image with a gradient descent over the unknown parameters of the measurement operator. However, this prior work does not leverage any learned priors for the measurement operator and does not provide any theoretical analysis.

**Our contributions.** *(1)* Our first contribution is in the use of learned deep denoisers for regularizing the measurement operators within PnP. While the idea of calibration within PnP was introduced in [43], denoisers were not used as priors for measurement operators. *(2)* Our second contribution is the application of BC-PnP as an efficient method for jointly estimating the unknown image and the measurement operator. While BC-RED was introduced in [21] as a block-coordinate variant of PnP, the method was used for solving non-blind inverse problems by using patch-based image denoisers. *(3)* Our third contribution is a new convergence theory for BC-PnP for the *sequential* and *random* block-selection strategies under approximate MMSE denoisers. Our analysis does *not* assume convex data-fidelity terms, which makes it compatible with blind inverse problems. Our analysis can be seen as an extension of [23] to block-coordinate updates and approximate MMSE denoisers. *(4)* Our fourth contribution is the implementation of BC-PnP using learned deep denoisers as priors for two distinct blind inverse problems: blind deblurring and auto-calibrated parallel MRI. Our code—which we share publicly—shows the potential of learning deep denoisers over measurement operators and using them for jointly estimating the uknown image and the uknown measurement operator.

## 3 Block Coordinate Plug-and-Play Method

We propose to efficiently solve blind inverse problems by using a block-coordinate PnP method, where each block represents one group of unknown variables (images, measurement operators, etc). The novelty of our work relative to [21] is in solving blind inverse problems by introducing learned priors on both the unknown image and the uknown measurement operator. Additionally, unlike [21], our work proposes a fully nonconvex formulation that is more applicable to blind inverse problems.

Consider the decomposition of a vector $\boldsymbol{x} \in \mathbb{R}^n$ into $b \geq 1$ blocks

$$\boldsymbol{x} = (\boldsymbol{x}_1, \cdots, \boldsymbol{x}_b) \in \mathbb{R}^{n_1} \times \cdots \times \mathbb{R}^{n_b} \quad \text{with} \quad n = n_1 + \cdots + n_b. \tag{3}$$

**Algorithm 1** Block Coordinate Plug-and-Play Method (BC-PnP)

---

1: **input:** initial value $\boldsymbol{x}^0 \in \mathbb{R}^n$, parameters $\boldsymbol{\sigma} \in \mathbb{R}_+^b$, and step-size $\gamma > 0$.
2: **for** $k = 1, 2, 3, \cdots$ **do**
3:     Choose an index $i_k \in \{1, \cdots, b\}$
4:     $\boldsymbol{x}^k \leftarrow \boldsymbol{x}^{k-1} - \gamma \mathbf{U}_{i_k} \mathsf{G}_{i_k}(\boldsymbol{x}^{k-1})$
       where $\mathsf{G}_i(\boldsymbol{x}) := \mathbf{U}_i^\mathsf{T} \mathsf{G}(\boldsymbol{x})$ with $\mathsf{G}(\boldsymbol{x}) := \frac{1}{\gamma}(\boldsymbol{x} - \mathsf{D}_{\boldsymbol{\sigma}}(\boldsymbol{x} - \gamma \nabla g(\boldsymbol{x})))$.
5: **end for**

---

For each $i \in \{1, \cdots, b\}$, we define a matrix $\mathbf{U}_i \in \mathbb{R}^{n \times n_i}$ that injects a vector in $\mathbb{R}^{n_i}$ into $\mathbb{R}^n$ and its transpose $\mathbf{U}_i^\mathsf{T}$ that extracts the $i$th block from a vector in $\mathbb{R}^n$. For any $\boldsymbol{x} \in \mathbb{R}^n$, we have

$$\boldsymbol{x} = \sum_{i=1}^b \mathbf{U}_i \boldsymbol{x}_i \quad \text{with} \quad \boldsymbol{x}_i = \mathbf{U}_i^\mathsf{T} \boldsymbol{x} \in \mathbb{R}^{n_i}, \quad i = 1, \cdots, b \quad \Leftrightarrow \quad \sum_{i=1}^b \mathbf{U}_i \mathbf{U}_i^\mathsf{T} = \mathbf{I}. \quad (4)$$

Note that (4) directly implies the norm preservation $\|\boldsymbol{x}\|_2^2 = \|\boldsymbol{x}_1\|_2^2 + \cdots + \|\boldsymbol{x}_b\|_2^2$ for any $\boldsymbol{x} \in \mathbb{R}^n$. We are interested in a block-coordinate algorithm that uses only a subset of operator outputs corresponding to coordinates in some block $i \in \{1, \cdots, b\}$. Hence, for an operator $\mathsf{G} : \mathbb{R}^n \to \mathbb{R}^n$, we define the block-coordinate operator $\mathsf{G}_i : \mathbb{R}^n \to \mathbb{R}^{n_i}$ as

$$\mathsf{G}_i(\boldsymbol{x}) := [\mathsf{G}(\boldsymbol{x})]_i = \mathbf{U}_i^\mathsf{T} \mathsf{G}(\boldsymbol{x}) \in \mathbb{R}^{n_i}, \quad \boldsymbol{x} \in \mathbb{R}^n. \quad (5)$$

We are in-particular interested in two operators: (a) the gradient $\nabla g(\boldsymbol{x}) = (\nabla_1 g(\boldsymbol{x}), \cdots, \nabla_b g(\boldsymbol{x}))$ of the data-fidelity term $g$ and (b) the denoiser $\mathsf{D}_{\boldsymbol{\sigma}}(\boldsymbol{x}) = (\mathsf{D}_{\sigma_1}(\boldsymbol{x}_1), \cdots, \mathsf{D}_{\sigma_b}(\boldsymbol{x}_b))$, where the vector $\boldsymbol{\sigma} = (\sigma_1, \cdots, \sigma_b) \in \mathbb{R}_+^b$ consists of parameters for controling the strength of each block denoiser. Note how the denoiser acts in a separable fashion across different blocks.

When $b = 1$, we have $\mathbf{U}_1 = \mathbf{U}_1^\mathsf{T} = \mathbf{I}$ and BC-PnP reduces to the conventional PnP-ISTA [23, 65]. When $b > 1$, we have at least two blocks with BC-PnP updating only one block at a time

$$\boldsymbol{x}_j^k = \begin{cases} \boldsymbol{x}_j^{k-1} & \text{when } j \neq i_k \\ \mathsf{D}_{\sigma_j}(\boldsymbol{x}_j^{k-1} - \gamma \nabla_j g(\boldsymbol{x}^{k-1})) & \text{when } j = i_k \end{cases}, \quad j \in \{1, \cdots, b\}. \quad (6)$$

As with any coordinate descent method (see [74] for a review), BC-PnP can be implemented using different block selection strategies. One common strategy is to simply update blocks sequentially as $i_k = 1 + \text{mod}(k - 1, b)$, where $\text{mod}(\cdot)$ denotes the modulo operator. An alternative is to proceed in epochs of $b$ consecutive iterations, where at the start of each epoch the set $\{1, \cdots, b\}$ is reshuffled, and $i_k$ is then selected consecutively from this ordered set. Finally, one can adopt a fully randomized strategy where indices $i_k$ are selected as i.i.d. random variables distributed uniformly over $\{1, \cdots, b\}$.

Throughout this work, we will assume that each denoiser $\mathsf{D}_{\sigma_i}$ is an *approximate* MMSE estimator for the following AWGN denoising problem

$$\boldsymbol{z}_i = \boldsymbol{x}_i + \boldsymbol{n}_i \quad \text{with} \quad \boldsymbol{x}_i \sim p_{\boldsymbol{x}_i}, \quad \boldsymbol{n}_i \sim \mathcal{N}(0, \sigma_i^2 \mathbf{I}), \quad (7)$$

where $i \in \{1, \cdots, b\}$ and $\boldsymbol{z}_i \in \mathbb{R}^{n_i}$. We rely only on an *approximation* of the MMSE estimator of $\boldsymbol{x}_i$ given $\boldsymbol{z}_i$, since the *exact* MMSE denoiser corresponds to the generally intractable posterior mean

$$\mathsf{D}_{\sigma_i}^*(\boldsymbol{z}_i) := \mathbb{E}[\boldsymbol{x}_i | \boldsymbol{z}_i] = \int_{\mathbb{R}^{n_i}} \boldsymbol{x} p_{\boldsymbol{x}_i | \boldsymbol{z}_i}(\boldsymbol{x} | \boldsymbol{z}_i) \, \mathrm{d}\boldsymbol{x}. \quad (8)$$

Approximate MMSE denoisers are a useful model for denoisers due to the use of the MSE loss

$$\mathcal{L}(\mathsf{D}_{\sigma_i}) = \mathbb{E}\left[\|\boldsymbol{x}_i - \mathsf{D}_{\sigma_i}(\boldsymbol{z}_i)\|_2^2\right] \quad (9)$$

for training deep denoisers, as well as the optimality of MMSE denoisers with respect to widely used image-quality metrics such as signal-to-noise ratio (SNR).

As a simple illustration of the generality of BC-PnP, consider $b = 2$ with the least-squares objective

$$g(\boldsymbol{x}) = \frac{1}{2}\|\boldsymbol{y} - \boldsymbol{A}(\boldsymbol{\theta})\boldsymbol{v}\|_2^2 \quad \text{with} \quad \boldsymbol{x} := (\boldsymbol{v}, \boldsymbol{\theta}), \quad (10)$$

where $\boldsymbol{v} \in \mathbb{R}^{n_1}$ denotes the unknown image and $\boldsymbol{\theta} \in \mathbb{R}^{n_2}$ denotes the unknown parameters of the measurement operator. BC-PnP can then be implemented by first pre-training a dedicated AWGN denoiser $\mathsf{D}_{\sigma_i}$ for each block $i$ and using it as a prior within Algorithm 1. It is also worth noting that the functions $g$ in (10) is nonconvex with respect to the variable $\boldsymbol{x} \in \mathbb{R}^n$. In the next section, we present the full convergence analysis of BC-PnP without any convexity assumptions on $g$ and nonexpansiveness assumptions on the denoiser $\mathsf{D}_{\boldsymbol{\sigma}}$.

## 4  Convergence Analysis of BC-PnP

In this section, we present two new theoretical convergence results for BC-PnP. We first discuss its convergence under the sequential updates and then under fully random updates. It is worth mentioning that BC-RED with fully random updates was theoretically analyzed in [21]. The novelty of our analysis here lies in that it allows for nonconvex functions $g$ and expansive denoisers $\mathsf{D}_{\sigma_i}$. The nonconvexity of $g$ is essential since most data-fidelity terms used for blind inverse problems are nonconvex. On the other hand, by allowing expansive $\mathsf{D}_{\sigma_i}$, our analysis avoids the need for the spectral normalization techniques that were previously suggested for PnP methods [21, 22].

In the following, we will denote as $\mathsf{D}_{\boldsymbol{\sigma}}^* := (\mathsf{D}_{\sigma_1}^*, \cdots, \mathsf{D}_{\sigma_b}^*)$ the exact MMSE denoiser in (8). Our analysis will require five assumptions that will serve as sufficient conditions for our theorems.

**Assumption 1.** *The blocks $\boldsymbol{x}_i$ are independent with non-degenerate priors $p_{\boldsymbol{x}_i}$ over $\mathbb{R}^{n_i}$.*

As a reminder, a probability distribution $p_{\boldsymbol{x}_i}$ is *degenerate* over $\mathbb{R}^{n_i}$, if it is supported on a space of lower dimensions than $n_i$. Assumption 1 is required for establishing an explicit link between the MMSE denoiser (8) and the following regularizer (see also [23, 66] for additional background)

$$h(\boldsymbol{x}) = \sum_{i=1}^{b} h_i(\boldsymbol{x}_i), \quad \boldsymbol{x} = (\boldsymbol{x}_1, \cdots, \boldsymbol{x}_n) \in \mathbb{R}^n, \tag{11}$$

where each function $h_i$ is defined as (see the derivation in Section D.2 of the supplement)

$$h_i(\boldsymbol{x}_i) := \begin{cases} -\frac{1}{2\gamma}\|\boldsymbol{x}_i - (\mathsf{D}_{\sigma_i}^*)^{-1}(\boldsymbol{x}_i)\|_2^2 + \frac{\sigma_i^2}{\gamma} h_{\sigma_i}((\mathsf{D}_{\sigma_i}^*)^{-1}(\boldsymbol{x}_i)) & \text{for } \boldsymbol{x}_i \in \mathsf{Im}(\mathsf{D}_{\sigma_i}^*) \\ +\infty & \text{for } \boldsymbol{x}_i \notin \mathsf{Im}(\mathsf{D}_{\sigma_i}^*), \end{cases} \tag{12}$$

where $\gamma > 0$ is the step size, $(\mathsf{D}_{\sigma_i}^*)^{-1} : \mathsf{Im}(\mathsf{D}_{\sigma_i}^*) \to \mathbb{R}^{n_i}$ is the inverse mapping, which is well defined and smooth over $\mathsf{Im}(\mathsf{D}_{\sigma_i}^*)$, and $h_{\sigma_i}(\cdot) := -\log(p_{\boldsymbol{z}_i}(\cdot))$, where $p_{\boldsymbol{z}_i}$ is the probability distribution over the AWGN corrupted observations (7). Note that the function $h_i$ is smooth for any $\boldsymbol{x}_i \in \mathsf{Im}(\mathsf{D}_{\sigma_i}^*)$, which is the consequence of the smoothness of both $(\mathsf{D}_{\sigma_i}^*)^{-1}$ and $h_{\sigma_i}$.

**Assumption 2.** *The function $g$ is continuously differentiable and $\nabla g$ is Lipschitz continuous with constant $L > 0$. Additionally, each block gradient $\nabla_i g$ is block Lipschitz continuous with constant $L_i > 0$. We define the largest block Lipschitz constant as $L_{\mathsf{max}} := \mathsf{max}\{L_1, \cdots, L_b\}$.*

Lipschitz continuity of the gradient $\nabla g$ is a standard assumption in the context of imaging inverse problems. Note that we always have the relationship $(L/b) \le L_{\mathsf{max}} \le L$ (see Section 3.2 in [74]).

**Assumption 3.** *The explicit data-fidelity term and the implicit regularizer are bounded from below*

$$\inf_{\boldsymbol{x} \in \mathbb{R}^n} g(\boldsymbol{x}) > -\infty, \quad \inf_{\boldsymbol{x} \in \mathbb{R}^n} h(\boldsymbol{x}) > -\infty. \tag{13}$$

Assumption 3 implies that there exists $f^* > -\infty$ such that $f(\boldsymbol{x}) \ge f^*$ for all $\boldsymbol{x} \in \mathbb{R}^n$. Since Assumptions 1-3 correspond to the standard assumptions used in the literature, they are broadly satisfied in the context of inverse problems.

Our analysis assumes that at every iteration, BC-PnP uses inexact MMSE denoisers on each block. While there are several ways to specify the nature of "inexactness," we consider the case where at every iteration $k$ of BC-PnP the distance of the output of $\mathsf{D}_{\sigma_i}$ to $\mathsf{D}_{\sigma_i}^*$ is bounded by a constant $\varepsilon_k$.

**Assumption 4.** *Each block denoiser $\mathsf{D}_{\sigma_i}$ in $\mathsf{D}_{\boldsymbol{\sigma}}$ satisfies*

$$\|\mathsf{D}_{\sigma_i}(\boldsymbol{z}_i^k) - \mathsf{D}_{\sigma_i}^*(\boldsymbol{z}_i^k)\|_2 \le \varepsilon_k, \quad i \in \{1, \cdots, b\}, \quad k = 1, 2, 3, \cdots,$$

*where $\mathsf{D}_{\sigma_i}^*$ is given in (8) and $\boldsymbol{z}_i^k = \boldsymbol{x}_i^{k-1} - \gamma \nabla_i g(\boldsymbol{x}^{k-1})$.*

For convenience, we will define quantities $\varepsilon^2 := \max\{\varepsilon_1^2, \varepsilon_2^2, \cdots\}$ and $\bar{\varepsilon}_t^2 := (1/t)\left(\varepsilon_1^2 + \cdots + \varepsilon_t^2\right)$ that correspond to the largest and the mean squared-distances between the inexact and exact denoisers. Assumption 4 states that the error of the approximate MMSE denoiser used for inference is bounded relative to the exact MMSE denoiser, which is reasonable when the approximate MMSE denoiser is a CNN trained to minimize the MSE.

It has been shown in the prior work [23, 66] that the function $h$ in (11) is infinitely continuously differentiable over $\mathsf{Im}(\mathsf{D}_{\boldsymbol{\sigma}}^*)$. Our analysis requires the extension of the region where $h$ is smooth to include the range of the approximate MMSE denoiser, which is the goal of our next assumption.

**Assumption 5.** *Each regularizer $h_i$ in (12) associated with the MMSE denoiser (8) is continuously differentiable and has a Lipschitz continuous gradient with constant $M_i > 0$ over the set*

$$\mathsf{Im}_\varepsilon(\mathsf{D}_{\sigma_i}^*) := \{\boldsymbol{x} \in \mathbb{R}^{n_i} : \|\boldsymbol{x} - \mathsf{D}_{\sigma_i}^*(\boldsymbol{z})\|_2 \leq \varepsilon, \ \boldsymbol{z} \in \mathbb{R}^{n_i}\}, \quad i \in \{1, \cdots, b\}.$$

We will define $M_{\mathsf{max}} := \max\{M_1, \cdots, M_b\}$ to be the largest Lipschitz constant and $\mathsf{Im}_\varepsilon(\mathsf{D}_{\boldsymbol{\sigma}}^*) := \{\boldsymbol{x} \in \mathbb{R}^n : \boldsymbol{x}_i \in \mathsf{Im}_\varepsilon(\mathsf{D}_{\sigma_i}^*), i \in \{1, \ldots, b\}\}$ to be the set over which $h$ is smooth. Assumption 5 expands the region where the regularizer associated with the exact MMSE denoiser is smooth by including the range of the approximate MMSE denoiser. For example, this assumption is automatically true when the exact and approximate MMSE denoisers have the same range, which is reasonable when the approximate MMSE denoiser is trained to imitate the exact one.

Our first theoretical result considers the sequential updates, where at each iteration, $i_k$ is selected as $i_k = 1 + \mathsf{mod}(k - 1, b)$ with $\mathsf{mod}(\cdot)$ being the modulo operator. We can then express any iterate $ib$ produced by BC-PnP for $i \geq 1$ as

$$\boldsymbol{x}^{ib} = (\boldsymbol{x}_1^{ib}, \cdots, \boldsymbol{x}_b^{ib}) = (\boldsymbol{x}_1^{(i-1)b+1}, \cdots, \boldsymbol{x}_b^{ib}).$$

Note that $\boldsymbol{x}^{ib} \in \mathsf{Im}_\varepsilon(\mathsf{D}_{\boldsymbol{\sigma}}^*)$ since each block is an output of the denoiser. We prove the following result.

**Theorem 1.** *Run BC-PnP under Assumptions 1-5 using the sequential block selection and the step $0 < \gamma < 1/L_{\mathsf{max}}$. Then, we have*

$$\min_{1 \leq i \leq t} \|\nabla f(\boldsymbol{x}^{ib})\|_2^2 \leq \frac{1}{t}\sum_{i=1}^{t} \|\nabla f(\boldsymbol{x}^{ib})\|_2^2 \leq \frac{C_1}{t}(f(\boldsymbol{x}^0) - f^*) + C_2\bar{\varepsilon}_{tb}^2,$$

*where $C_1 > 0$ and $C_2 > 0$ are iteration independent constants. If additionally the sequence of error terms $\{\varepsilon_i\}_{i \geq 1}$ is square-summable, we have that $\nabla f(\boldsymbol{x}^{tb}) \to \boldsymbol{0}$ as $t \to 0$.*

Our second theoretical result considers fully random updates, where at each iteration, $i_k$ is selected as i.i.d. random variables distributed over $\{1, \cdots, b\}$. In this setting, we analyze the convergence of BC-PnP in terms of the sequence $\{\mathsf{G}(\boldsymbol{x}^k)\}_{k \geq 0}$. Note that it is straightforward to verify that $\mathsf{Zer}(\mathsf{G}) = \mathsf{Zer}(\nabla f)$, which makes this analysis meaningful. We prove the following result.

**Theorem 2.** *Run BC-PnP under Assumptions 1-5 using the random i.i.d. block selection and the step $0 < \gamma < 1/L_{\mathsf{max}}$. Then, we have*

$$\min_{1 \leq k \leq t} \mathbb{E}\left[\|\mathsf{G}(\boldsymbol{x}^{k-1})\|_2^2\right] \leq \mathbb{E}\left[\frac{1}{t}\sum_{k=1}^{t} \|\mathsf{G}(\boldsymbol{x}^{k-1})\|_2^2\right] \leq \frac{D_1}{t}(f(\boldsymbol{x}^0) - f^*) + D_2\bar{\varepsilon}_t^2,$$

*where $D_1 > 0$ and $D_2 > 0$ are iteration independent constants. If additionally the sequence of error terms $\{\varepsilon_i\}_{i \geq 1}$ is square-summable, we have that $\mathsf{G}(\boldsymbol{x}^t) \xrightarrow{\text{a.s.}} \boldsymbol{0}$ as $t \to \infty$.*

The expressions for the constants in Theorems 1 and 2 are given in the proofs. The theorems show that if the sequence of approximation errors is square-summable, BC-PnP asymptotically achieves a stationary point of $f$. On the other hand, if the sequence of approximation errors is not square-summable, the convergence is only up to an error term that depends on the average of the squared approximation errors. Both theorems can thus be viewed as more flexible alternatives for the convergence analysis in [21]. It is also worth mentioning that the theorems are interesting even when the denoiser errors are not square-summable, since they provide explicit error bounds on convergence. While the analysis in [21] assumes convex $g$ and nonexpansive $\mathsf{D}_{\boldsymbol{\sigma}}$, the analysis here does not require these two assumptions. It instead views the denoiser $\mathsf{D}_{\boldsymbol{\sigma}}$ as an *approximation* to the MMSE estimator $\mathsf{D}_{\boldsymbol{\sigma}}^*$, where the approximation error is bounded by $\varepsilon_k$ at every iteration of BC-PnP. This view is compatible with denoisers trained to minimize the MSE loss (9).

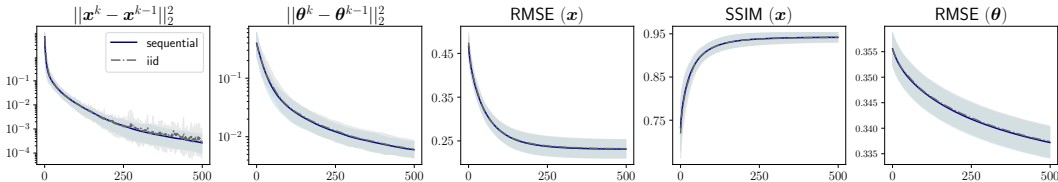

Figure 1: *Illustration of the BC-PnP convergence using the sequential and random i.i.d. block selection rules on CS-PMRI with the sampling factor $R = 8$. Leftmost two plots: Evolution of the distance between two consecutive image and CSM iterates. Rightmost three plots: Evolution of the RMSE and SSIM metrics relative to the true solutions across BC-PnP iterations. Note how both block selection rules lead to a nearly identical convergence behaviour of BC-PnP in this experiment.*

## 5   Numerical Validation

We numerically validate BC-PnP on two blind inverse problems: (a) *compressed sensing parallel MRI (CS-PMRI)* with automatic coil sensitivity map (CSM) estimation and (b) *blind image deblurring*. We adopt the traditional $\ell_2$-norm loss in (10) as the data-fidelity term for both problems. We will use $x$ to denote the unknown image and $\theta$ to denote the unknown parameters of the measurement operator. We use the relative root mean squared error (RMSE) and structural similarity index (SSIM) as quantitative metrics to evaluate the performance.

We experimented with several ablated variants of BC-PnP, including PnP, PnP-GD$_\theta$, and PnP-oracle$_\theta$. PnP and PnP-oracle$_\theta$ denote basic variants of PnP that use pre-estimated and ground truth measurement operators, respectively. PnP-GD$_\theta$ is a variant of PnP based on [43], where $\theta$ is estimated without any DL prior. It is worth noting that PnP-oracle$_\theta$ is provided as an idealized reference in our experiment. As discussed in the following subsections, we also compare BC-PnP against several widely-used baseline methods specific to CS-PMRI and blind image deblurring.

### 5.1   Compressed Sensing Parallel MRI

The measurement operator of CS-PMRI consists of complex measurement operators $\boldsymbol{A}(\boldsymbol{\theta}) \in \mathbb{C}^{m \times n}$ that depend on unknown CSMs $\{\boldsymbol{\theta}_i\}$ in $\mathbb{C}^n$. Each sub-measurement operator can be parameterized as $\boldsymbol{A}_i(\boldsymbol{\theta}_i) = \boldsymbol{P} \boldsymbol{F} \mathrm{diag}(\boldsymbol{\theta}_i)$, where $\boldsymbol{F}$ is the Fourier transform, $\boldsymbol{P} \in \mathbb{R}^{m \times n}$ is the sampling operator, and $\mathrm{diag}(\boldsymbol{\theta}_i)$ forms a matrix by placing $\boldsymbol{\theta}_i$ on its diagonal. We used T2-weighted MR brain acquisitions of 165 subjects obtained from the validation set of the fastMRI dataset [75] as the the fully sampled measurement for simulating measurements. We obtained reference $\boldsymbol{\theta}_i$ from the fully sampled measurements using ESPIRiT [76]. These 165 subjects were split into 145, 10, and 10 for training, validation, and testing, respectively. BC-PnP and baseline methods were tested on 10 2D slices, randomly selected from the testing subjects. We followed [75] to retrospectively undersample the fully sampled data using 1D Cartesian equispaced sampling masks with 10% auto-calibration signal (ACS) [76] lines. We conducted our experiments for acceleration factors $R = 6$ and 8. We adopted DRUNet [12] as the architectures of $\mathsf{D}_{\boldsymbol{\sigma}}$ for training both the image and CSM denoisers. BC-PnP and its ablated variants are initialized using CSMs $\boldsymbol{\theta}_0$ pre-estimated using ESPIRiT [76] and images $x_0 \leftarrow \boldsymbol{A}(\boldsymbol{\theta}_0)^{\mathsf{H}} \boldsymbol{y}$, where $\boldsymbol{A}^{\mathsf{H}}$ denotes the Hermitian transpose of $\boldsymbol{A}$.

We considered several baseline methods, including ENLIVE [35], ESPIRiT-TV [76], Unet [77], and ISTANet+ [51]. ENLIVE is an iterative algorithm that jointly estimates images and coil sensitivity profiles. ESPIRiT-TV is an iterative algorithm that applies TV reconstruction method in (1). Unet is trained to map raw measurements to desired ground truth without the knowledge of measurement operator. ISTANet+ denotes a widely-used DU architecture. We tested ESPIRiT-TV and ISTANet+ using CSMs pre-estimated using ESPIRiT.

Figure 1 illustrates the convergence behaviour of BC-PnP on the test set for the acceleration factor $R = 8$. Figure 2 illustrates reconstruction results for the acceleration factor $R = 6$. Table 1 summarizes the quantitative evaluation of BC-PnP relative to other PnP variants and the baseline methods. These results show that joint estimation can lead to significant improvements and that BC-PnP can perform as well as the idealized PnP-oracle$_\theta$ that knowns the true measurement operator.

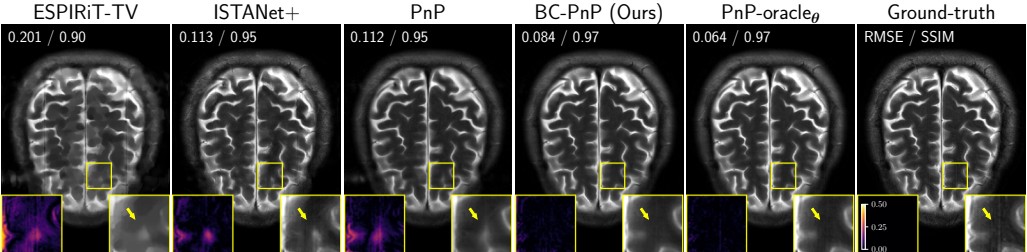

Figure 2: *Illustration of results from several well-known methods on CS-PMRI with the sampling factor $R = 6$. The quantities in the top-left corner of each image provide RMSE and SSIM values for each method. The squares at the bottom of each image visualize the error and the corresponding zoomed area in the image. Note how BC-PnP using a deep denoiser on the unknown CSMs outperforms uncalibrated PnP and matches PnP-oracle$_\theta$ that knows the true CSMs.*

Table 1: RMSE and SSIM performance of several methods on CS-PMRI. The table highlights the **best** and second best results. The *Calibration* column highlights methods specifically designed to solve the blind inverse problem. Note how the use of a DL prior over the measurement operator enables BC-PnP to outperform PnP and PnP-GD$_\theta$ and approach the performance of the oracle algorithm.

| Method | Calibration (Y/N) | $R = 6$ | | | $R = 8$ | | |
|---|---|---|---|---|---|---|---|
| | | RMSE$_x \downarrow$ | SSIM$_x \uparrow$ | RMSE$_\theta \downarrow$ | RMSE$_x \downarrow$ | SSIM$_x \uparrow$ | RMSE$_\theta \downarrow$ |
| ENLIVE [35] | ✓ | 0.371 | 0.763 | — | 0.419 | 0.730 | — |
| ESPIRiT-TV [76] | ✓ | 0.218 | 0.884 | 0.256 | 0.361 | 0.818 | 0.356 |
| Unet [77] | ✗ | 0.218 | 0.904 | — | 0.195 | 0.907 | — |
| ISTANet+ [51] | ✗ | 0.110 | 0.946 | — | 0.140 | 0.928 | — |
| PnP | ✗ | 0.111 | 0.950 | 0.256 | 0.171 | 0.924 | 0.356 |
| PnP-GD$_\theta$ [43] | ✓ | 0.116 | 0.950 | 0.254 | 0.163 | 0.926 | 0.355 |
| BC-PnP (Ours) | ✓ | **0.091** | **0.961** | **0.247** | **0.122** | **0.946** | **0.337** |
| PnP-oracle$_\theta$* | ✗ | 0.069 | 0.969 | 0.000 | 0.082 | 0.962 | 0.000 |

* not available in practice for blind inverse problems.

## 5.2 Blind Image Deblurring

The measurement operator in blind image deblurring can be modeled as $A(\theta)x = \theta * x$, where $\theta$ is the unknown blur kernel, $x$ is the unknown image, and $*$ is the convolution. We randomly selected 10 testing ground truth image from CBSD68 [80] dataset. We generated $25 \times 25$ Gaussian kernels with $\sigma = 10^1$. We tested the algorithms on 2 Gaussian kernels. We used a pre-trained image denoiser, as in the experimental setting of [12]. The kernel denoiser was trained on 10,000 generated kernels at several noise levels. We adopted DnCNN with 17 layers as the architectures of $D_\sigma$ for training kernel denoisers. BC-PnP and its ablated variants are initialized with the blur kernels $\theta_0$ pre-estimated using Pan-DCP [39] and images $x_0 \leftarrow A(\theta_0)^\mathsf{T} y$.

We compared BC-PnP against several baseline methods, including Pan-DCP [39], SelfDeblur [58], DeblurGAN [78], USRNet [79]. Pan-DCP is an optimization-based method that jointly estimates image and blur kernel. SelfDeblur trains two deep image priors (DIP) [81] to jointly estimate the blur kernel and the image. DeblurGAN is a supervised learning-based method that lacks the capability for kernel estimation, but can reconstruct images via direct inference. USRNet is a DU baseline that was tested using blur kernel estimated from [39]. The results of DeblurGAN and USRNet are obtained by running the published code with the pre-trained weights.

Figure 3 illustrates the reconstruction results with a Gaussian kernel. Figure 3 demonstrates that BC-PnP can reconstruct the fine details of the snake skin, as highlighted by the white arrows, while both Pan-DCP and PnP produce smoother reconstructions. Additionally, BC-PnP generates a more accurate blur kernel compared to the ground truth kernel, whereas Pan-DCP and SelfDeblur yield blur kernels with artifacts. Table 2 presents the quantitative evaluation of the reconstruction results using

---

[1] We used `github.com/shangqigao/BayeSR` for generating the kernels.

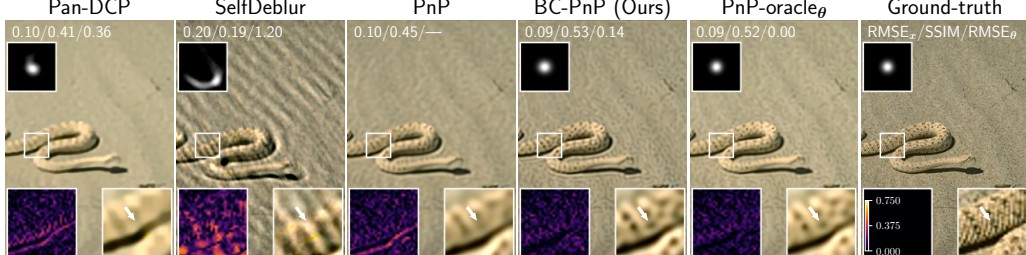

Figure 3: *Illustration of results from several well-known methods on blind image deblurring with the Gaussian kernel. The squares at the top of each image show the estimated kernels. The quantities in the top-left corner of each image provide RMSE and SSIM values for each method. The squares at the bottom of each image highlight the error and the corresponding zoomed image region. Note how the BC-PnP using a deep denoiser on the unknown kernel significantly outperforms the traditional PnP method and matches the performance of the oracle PnP method that knows the true blur kernel. Note also the effectiveness of BC-PnP for estimating the unknown blur kernel.*

Table 2: Quantitative evaluation of BC-PnP in blind image deblurring. We highlighted the best and second best results, respectively. The *Calibration* column highlights methods specifically designed to solve the blind inverse problem. Note how the use of a prior over the measurement operator enables BC-PnP to nearly match the performance of the oracle algorithm.

| Method | Calibration (Y/N) | RMSE$_x$ ↓ | SSIM$_x$ ↑ | RMSE$_\theta$ ↓ | RMSE$_x$ ↓ | SSIM$_x$ ↑ | RMSE$_\theta$ ↓ |
|---|---|---|---|---|---|---|---|
| Pan-DCP [39] | ✓ | 0.087 | 0.835 | 0.283 | 0.114 | 0.733 | 0.246 |
| SelfDeblur [58] | ✓ | 0.219 | 0.495 | 0.775 | 0.176 | 0.553 | 0.831 |
| DeblurGAN [78] | ✗ | 0.090 | 0.823 | — | 0.118 | 0.716 | — |
| USRNet [79] | ✗ | 0.106 | 0.855 | — | 0.114 | 0.769 | — |
| PnP | ✗ | 0.082 | 0.857 | 0.283 | 0.106 | 0.763 | 0.246 |
| PnP-GD$_\theta$ [43] | ✓ | 0.082 | 0.857 | 0.283 | 0.108 | 0.767 | 0.246 |
| BC-PnP (Ours) | ✓ | **0.055** | **0.921** | **0.097** | **0.098** | **0.794** | **0.107** |
| PnP-oracle$_\theta$* | ✗ | 0.051 | 0.929 | 0.000 | 0.088 | 0.817 | 0.000 |

* not available in practice for blind inverse problems.

a Gaussian kernel, indicating that BC-PnP outperforms the baseline methods and nearly matches the SSIM and RMSE values of PnP-oracle$_\theta$ that is based on the ground truth blur kernel.

## 6    Conclusion

The work presented in this paper proposes a new BC-PnP method for jointly estimating unknown images and unknown measurement operators in blind inverse problems, presents its theoretical analysis in terms of convergence and accuracy, and applies the method to two well-known blind inverse problems. The proposed method and its theoretical analysis extend the recent work on PnP by introducing a learned prior on the unknown measurement operator, dropping the convexity assumptions on the data-fidelity term, and nonexpansiveness assumptions on the denoiser. The numerical validation of BC-PnP shows the improvements due to the use of learned priors on the measurement operator and the ability of the method to match the performance of the oracle PnP method that knows the true measurement operator. One conclusion of this work is the potential effectiveness of PnP for solving inverse problems where the unknown quantities are not only images.

## Limitations

The work presented in this paper comes with several limitations. The proposed BC-PnP method is based on PnP, which means that its performance is inherently limited by the use of AWGN denoisers as priors. While denoisers provide a convenient, principled, and flexible mechanism to specify priors,

they are inherently self-supervised and their empirical performance can thus be suboptimal compared to priors trained in a supervised fashion for a specific inverse problem. PnP running over many iterations can also have higher computational complexity compared to some end-to-end alternatives, such as DU with a small number of steps. Our analysis is based on the assumption that the denoiser used for inference computes an approximation of the true MMSE denoiser. While this assumption is reasonable for deep denoisers trained using the MSE loss, it is not directly applicable to denoisers trained using other common loss functions, such as the $\ell_1$-norm or SSIM. Finally, as is common with most theoretical work, our analysis only holds when our assumptions are satisfied, which might limit its applicability in practice. Our future work will investigate ways to improve on the results presented here by exploring new PnP strategies for relaxing assumptions for convergence, considering end-to-end trained variants of BC-PnP based on DEQ [53, 54], and exploring BC-PnP using explicit regularizers [26–28].

## Broader Impact

The expected impact of this work is in the area of imaging inverse problems with potential applications to computational imaging. There is a growing interest in computational imaging to leverage pre-trained deep models for estimating the unknown image as well as the unknown parameters of the imaging system. The ability to better address such problems can lead to new imaging tools for biomedical and scientific studies. While novel DL methods, such as the proposed BC-PnP approach, have the potential to enable new technological capabilities, they also come with a downside of being more complex and requiring higher-levels of technical sophistication. While our aim is to positively contribute to humanity, one can unfortunately envisage nonethical use of imaging technology.

## Acknowledgments and Disclosure of Funding

Research presented in this article was supported in part by the NSF CCF-2043134. This work is also supported by the NIH R01EB032713, RF1AG082030, RF1NS116565, and R21NS127425.

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
