# OpenReview forum: "Block Coordinate Plug-and-Play Methods for Blind Inverse Problems"
_NeurIPS.cc/2023/Conference — NeurIPS 2023 poster_

### Official Review · Reviewer_8yHJ · 2023-07-02

**Soundness:** 3 good
**Presentation:** 2 fair
**Contribution:** 2 fair
**Rating:** 4
**Confidence:** 4

**Summary:**

The paper proposes a block-coordinate optimization strategy for plug-and-play (PnP) methods to solve blind inverse problems in imaging. Empirical validation is performed with blind deblurring and parallel CS-MRI. Theoretical study on the convergence of the proposed BC-PnP was extended from the theoretical result of PnP with MMSE denoisers.

**Strengths:**

1. To the best of my knowledge, this is one of the first works to extend PnP methods to blind inverse problems

2. Theory is sound and proves proper convergence properties of the proposed method.

3. The method seems to be robust and generally applicable to inverse problems arising in imaging, as validated in the experiments.

**Weaknesses:**

1. I do not find much novelty in the theory of the paper. While the components are all sound, both components of the extension (extending to block coordinate update and extending to denoisers that are optimal only up to a constant error) seems to be quite trivial. I would appreciate it if the authors could elaborate on this point.

2. BC-PnP seems to rely quite heavily on initialization for the imaging operators (i.e. sensitivity maps, blur kernels) that are close to ground truth. Such initialization strategy is not always possible, and for many cases even when it is possible, is often computationally demanding. Does BC-PnP converge properly also for cases when there is no such initialization strategy? Adding on this point, it might not be fair to compare other blind inverse problem solver that do not leverage any initialization strategy.

3. For blind deblurring, it seems that the authors only tested on Gaussian blur kernels, which are known to be relatively easy to fit. Does the method scale to more complex blur kernels, such as in [1,2]?


**Referneces**

[1] Levin, Anat, et al. "Understanding and evaluating blind deconvolution algorithms." 2009 IEEE conference on computer vision and pattern recognition. IEEE, 2009.

[2] Chung, Hyungjin, et al. "Parallel diffusion models of operator and image for blind inverse problems." Proceedings of the IEEE/CVF Conference on Computer Vision and Pattern Recognition. 2023.

**Questions:**

Please see weaknesses

**Limitations:**

Yes.

---

> ### Author Rebuttal · Authors · 2023-08-08
>
> Thank you for your valuable feedback and positive comments on our work. Please see below for our point-by-point responses to your comments. The PDF with additional simulations is available within the Author Rebuttal panel above.
>
> **Novelty in the theory:**
> We respectfully disagree with the reviewer's evaluation of the novelty in our theory. On the contrary, we believe that our work presents one of the most compelling theoretical and conceptual extensions of PnP to date. Our theory extends the existing work on PnP by considering (**a**) non-convex data fidelity terms, (**b**) expansive-inexact denoisers, and (**c**) partial processing of data. Our theory addresses several common misconceptions about PnP in the community, including that (**a**) PnP requires convex data fidelity terms; (**b**) CNN priors in PnP require spectral normalization to ensure convergence; and (**c**) effectiveness of learned denoisers as priors is limited to images.
>
> **Commentary on initialization:**
> Indeed, initialization is a deep and important topic in the context of nonconvex optimization. Since BC-PnP is designed for nonconvex blind inverse problems, it certainly benefits from the availability of a good initialization method. Prompted by your comment, we ran additional simulations on CS-PMRI to test the performance of BC-PnP with another initialization strategy. In the new setting, we used the inverse Fourier transform of the central low-frequency k-space data (termed **Low-K**) as an initialization strategy for BC-PnP. This new Low-K method is known to be less optimal compared to the ESPRiT method used in the main paper, but offers significantly reduced computational complexity. We report the results in **Table 5** and **Figure 2** in the attached PDF. We also report convergence of BC-PnP under the Low-K initialization in **Figure 3** of the attached PDF. Overall, these results show that (**a**) while the performance of BC-PnP is slightly inferior when initialized via Low-K compared to ESPRiT, it still maintains its superior performance over the baseline methods, and (**b**) the convergence behavior of BC-PnP is similar for both initialization strategies.
>
> **Blind deblurring beyond Gaussian blur:**
> Prompted by your comment, we compared BC-PnP in blind image deblurring with motion kernels. We followed `github.com/LeviBorodenko/motionblur` to simulate motion kernels for training and testing. We report results of BC-PnP and other methods in **Table 6** in the attached pdf. The results show that BC-PnP can outperform PnP by using a prior over the measurement operator. It is also worth noting that given the short duration of the rebuttal window, these results are preliminary. Nevertheless, we are confident that BC-PnP's performance can be further improved through fine-tuning of hyperparameters.

---

> > ### Comment · Reviewer_8yHJ · 2023-08-10
> >
> > Thank you for your response. Overall, I am satisfied with the rebuttal.
> >
> > 1. After reading the response and going through other reviewers' comments, I believe that my concerns with the shortcomings of the theory may have been due to my lack of understanding.
> >
> > 2. The experiment on the initialization for CS-MRI indeed shows that BC-PnP will work under other less-close-to-the-ground-truth initializations. I am still curious if this will scale to motion deblur problems even when the kernel is initialized from e.g. Gaussian. That said, I do understand that a week is a short period of time to conduct all the experiments.
> >
> > 3. I am still a bit skeptical about the performance of motion blind deblurring, as the metrics seem quite similar to Pan-DCP, and there is no figure to compare the results.

---

> > > ### Author Response · Authors · 2023-08-14
> > >
> > > We thank the reviewer for acknowledging our rebuttal and for sharing additional thoughts. We hope that the reviewer will consider updating their ratings to reflect the rebuttal.
> > >
> > > 1. We appreciate the reviewer’s re-evaluation of their views on our theory. We are indeed very pleased to share our theoretical results with the community.
> > > 2. Our current empirical results show that BC-PnP can achieve excellent performance given realistic initialization strategies for MRI and Gaussian deblur. It would indeed be interesting to look deeper into the matter of initialization in the context of motion deblur, which we will have to leave for future work. However, our preliminary results indicate that the current method already provides an excellent starting point for this direction.
> > > 3. While our numerical results on motion deblur are very preliminary, they already indicate that BC-PnP can do well, even with very limited time for fine-tuning. We did not include visual illustrations for motion deblur purely due to space limitations in the rebuttal document. However, we can say that visually BC-PnP is competitive with other methods.

---

### Official Review · Reviewer_7kUv · 2023-07-04

**Soundness:** 3 good
**Presentation:** 3 good
**Contribution:** 3 good
**Rating:** 6
**Confidence:** 4

**Summary:**

This paper presents a new plug-and-play algorithm for solving blind inverse problems, i.e., where the forward sensing operator is not fully known. The method leverages pretrained denoisers for both the signals and the unknown parameters of the sensing model. A theoretical analysis of the proposed algorithm is presented, showing that under certain conditions on the denoisers and data fidelity functions, the algorithm is guaranteed to converge to a local minimizer.

**Strengths:**

- The paper presents a novel plug-and-play algorithm for blind inverse problems, with some theoretical guarantees under relatively mild assumptions.

- The experimental results show a good performance with respect to other competing methods in two different blind inverse problems.

**Weaknesses:**

I find it a bit surprising that the PnP method outperfoms other unrolled algorithms trained for a specific inverse problem. Even the authors express that "While denoisers provide a convenient, principled, and flexible mechanism to specify priors, they are inherently self-supervised and their empirical performance can thus be suboptimal compared to priors trained in a supervised fashion for a specific inverse problem"
I wonder if the comparison presented in the paper is fair, do the competing methods use architectures that have a similar expressive power than the DRUNet denoiser used for PnP? Moreover, there exist end-to-end architectures that can estimate both the forward operator parameters and the underlying image (e.g. "Deep algorithm unrolling for blind image deblurring" by Li et al.), which haven't been evaluated here.

**Questions:**

- What is the performance of the proposed method on the blind deblurring problem with non-isotropic kernels? It would be good to show an example in this setting.

**Limitations:**

The paper discusses some of the limitations of the proposed method. However, there is no discussion related to the inherent large computational complexity of PnP methods, that require many more iterations than unrolled networks to reach convergence.

---

> ### Author Rebuttal · Authors · 2023-08-08
>
> Thank you for your valuable feedback and positive comments on our work. Please see below for our point-by-point responses to your comments. The PDF with additional simulations is available within the Author Rebuttal panel above.
>
> **Commentary on deep unrolling:**
> We fully agree with the reviewer that in non-blind settings deep unrolling (**DU**) methods are expected to outperform PnP. Prompted by your comments, we tested this hypothesis using the DU method USRNet on image deblurring. We present the results in **Table 3** of the attached PDF. Note how USRNet-oracle that uses the true blur kernel outperforms PnP-oracle. However, PnP outperforms USRNet when the kernel used for inference is inaccurate. Since the prior in PnP is independent from the measurement operator, it is not surprising that PnP is more robust to inaccuracies in the measurement operator.
>
> Prompted by your comment, we compared BC-PnP against DUBLID (“Deep algorithm unrolling for blind image deblurring" by Li et al, 2019). We report the results in **Table 4** of the attached PDF. The results show that BC-PnP can quantitatively outperform DUBLID in blind image deblurring. Note that the blur kernels predicted by DUBLID differ in size compared to those used in our experiments. As a result, we excluded the quantitative evaluation of the predicted kernel from the table. We will cite DUBLID in the revised paper.
>
> **Blind deblurring beyond Gaussian blur:**
> Prompted by your comment, we compared BC-PnP in blind image deblurring with motion kernels. We followed `github.com/LeviBorodenko/motionblur` to simulate motion kernels for training and testing. We report results of BC-PnP and other methods in **Table 6** in the attached PDF. The results show that BC-PnP can outperform PnP by using a prior over the measurement operator. It is also worth noting that given the short duration of the rebuttal window, these results are preliminary. Nevertheless, we are confident that BC-PnP's performance can be further improved through fine-tuning of hyperparameters.
>
> Computational cost:
> We agree with the reviewer that PnP running over many iterations can have higher computational complexity than unrolled networks with a small number of steps. We will incorporate this perspective into the limitations section of the final manuscript.

---

> > ### Comment · Reviewer_7kUv · 2023-08-14
> >
> > Thanks for answering my comments, I am satisfied by the rebuttal and I will raise my score accordingly.
> >
> > I just have a concern regarding the comparison with DUBLID: the comparison seems to use an isotropic kernel, how would the algorithms compare for non-isotropic ones (e.g., the one in Table 6)?

---

> > > ### Author Response · Authors · 2023-08-17
> > >
> > > We thank the reviewer for raising the score and for sharing additional thoughts.
> > >
> > > Prompted by the additional comment, we tested DUBLID on the same motion kernel depicted in Figure 6 of the attached pdf, and compared its performance with BC-PnP. The results are as follows:
> > > | Method | NRMSE | SSIM  |
> > > |--            |--            |-- |
> > > | DUBLID | 0.183 | 0.606 |
> > > | BC-PnP | 0.123 | 0.741 |
> > > Note how BC-PnP outperforms DUBLID quantitatively. While it is not possible for us to show visual results in the discussion section, we can say that visually BC-PnP is better than DUBLID.

---

### Official Review · Reviewer_NG5f · 2023-07-05

**Soundness:** 3 good
**Presentation:** 3 good
**Contribution:** 3 good
**Rating:** 7
**Confidence:** 3

**Summary:**

The paper generalizes the plug-and-play framework from single-block to a multi-block case by embedding unknown inverse operators. It provides convergence analysis for the proposed method and conducted various experiments on two imaging applications, i.e., parallel MRI and blind image deblurring. Numerical results are sufficiently convincing to show that the proposed method works better than the other comparable PnP methods in the blind measurement operator case.

**Strengths:**

The paper reads well and is well-organized from theoretical discussions to numerical experiments. The proposed method is original with clear motivation of unknown inverse operators from the realistic setting. Thus, it may also shed some light on other imaging applications.

**Weaknesses:**

1. The computational cost of this framework is not analyzed or even mentioned. Likewise, the running times for all the experimental results could be listed and compared. Some other end-to-end methods may also be tested.
2. Some minor notational confusion exists, e.g., lines 198-199.
3. Theorems 1-2 require square-summable condition for the error sequences to ensure the convergence of gradients. Further discussions may be added about how this condition can be satisfied or checked in practice.

**Questions:**

In all the numerical experiments, are the assumptions 1-5 and conditions in Theorems 1-2 verified or checked? If yes, the details could be included in the supplement.

**Limitations:**

The authors have listed the limitations and provided some potential solutions in future works.

---

> ### Author Rebuttal · Authors · 2023-08-08
>
> Thank you for your valuable feedback and positive comments on our work. Please see below for our point-by-point responses to your comments. The PDF with additional simulations is available within the Author Rebuttal panel above.
>
> **Computational cost:** Prompted by your comment, we computed running times of BC-PnP and PnP for both blind image deblurring and CS-PMRI. The results are presented in **Table 2** of the attached PDF and will be included in the revised supplement. As expected, BC-PnP is slower than traditional PnP due to the need for updating the parameters of the measurement operators. The computational overhead of BC-PnP over PnP depends on the size of the measurement operator. For instance, in the context of blind image deblurring, where the variables of the measurement operators are 25 x 25 kernels, BC-PnP's computational overhead is about 7% over the traditional PnP. We will publicly release our code, which should enable others to easily compare the efficiency and performance of BC-PnP relative to other methods, including any specific end-to-end methods.
>
> **Clarifying the notations:** We will clarify the notation for modulo operator and its use in lines 198-199 in the revised paper.
>
> **Conditions for Theorems 1-2:** We agree with the reviewer that the paper would benefit from a more careful discussion of our assumptions. We will include the following discussion in the main paper after presenting our theory. Since Assumptions 1-3 correspond to the standard assumptions used in the literature, they are broadly satisfied in the context of inverse problems. Assumption 4 was introduced by us and states that the error of the approximate MMSE denoiser used for inference is bounded relative to the exact MMSE denoiser, which is reasonable when the approximate MMSE denoiser is a CNN trained to minimize the MSE. Assumption 5 is another assumption introduced by us to expand the region where the regularizer associated with the exact MMSE denoiser is smooth by including the range of the approximate MMSE denoiser. For example, this assumption is automatically true when the exact and approximate MMSE denoisers have the same range, which is reasonable when the approximate MMSE denoiser is trained to imitate the exact one. It is also worth mentioning that Theorems 1-2 are interesting even when the denoiser errors are **not** square-summable, since they provide explicit error bounds on convergence.

---

### Official Review · Reviewer_tTXT · 2023-07-08

**Soundness:** 4 excellent
**Presentation:** 4 excellent
**Contribution:** 4 excellent
**Rating:** 7
**Confidence:** 5

**Summary:**

Develops a plug-and-play technique for blind (unknown forward model) imaging inverse problems. Key idea is to use two denoising algorithms: One as a prior on the image and one as a prior model on the measurement operator (e.g., blur kernel). Also analyzes the convergence of the algorithm. Successfully applies (in simulation) the proposed algorithm to MRI with unknown coil sensitive maps and deblurring with unknown blur kernel.


**Strengths:**

Investigates important and understudied problem

Proposed technique is well-justified and seemingly effective

I like the paper's approach to framing the blind image reconstruction problem (equation (10)).

Easy to read

Proposed method seems generally applicable to lots of problems and opens up and opportunity for lots of follow-up work

Code will be shared publicly

**Weaknesses:**

Lacks comparisons with closest competitors. Particulary [60].

Missing related work: There's a line of work on PnP for holography with unknown phase errors (e.g., [A]) that may be worth mentioning.

[A] Pellizzari, Casey J., Mark F. Spencer, and Charles A. Bouman. "Coherent plug-and-play: digital holographic imaging through atmospheric turbulence using model-based iterative reconstruction and convolutional neural networks." IEEE Transactions on Computational Imaging 6 (2020): 1607-1621.


**Questions:**

Diffusion models are essentially denoisers by a different name, therefore a comparison with [60] seems obligatory. They are essentially solving the same problem (though the current method has more theory). How do they compare?


**Limitations:**

Adequately discussed

---

> ### Author Rebuttal · Authors · 2023-08-08
>
> Thank you for your valuable feedback and positive comments on our work. Please see below for our point-by-point responses to your comments. The PDF with additional simulations is available within the Author Rebuttal panel above.
>
> **Comparison with [60]:**
> Prompted by your comment, we ran simulations using **BlindDPS** from [60] on the blind deconvolution problem considered in our paper. We report the quantitative and visual results in **Table 1** and **Figure 1** in the PDF attached to our response, respectively. Note how BlindDPS achieves the SSIM value of 0.734, while BC-PnP achieves that of 0.921. Based on the other results reported on DPS, we are confident that BC-PnP is competitive with BlindDPS in terms of traditional image quality metrics such as PSNR or SSIM, while BlindDPS can do better on perceptual metrics such as FID. For example, Table 7 in the original DPS paper (`arXiv:2209.14687`) reports the superior performance of the traditional PnP-ADMM over DPS on Gaussian deblurring. It is also worth highlighting that there is value in doing more careful comparisons between BlindDPS and BC-PnP that include possible finetuning/retraining of models for specific settings (*e.g.*, types and sizes of blur kernels), which would be infeasible within this rebuttal window. Nonetheless, we will publicly release our code, which will enable easy comparisons and reproducibility of our results in the future.
>
> **Citing related work:**
> Thank you for sharing [A], which we will cite in the final paper.

---

### Author Rebuttal · Authors · 2023-08-08

Thank you all for providing us with valuable feedback. We provide detailed answers to all the comments below. To better address some of them, we ran additional simulations, albeit within the time constraints of the limited rebuttal window. The PDF file attached to this response reports the results of these simulations. We will update the paper and the supplementary material for the camera ready submission based on our responses.

---

### Author Response · Authors · 2023-08-19
**Discussion Period Response to All Reviewers and Area Chairs**

Thank you all again for reviewing our work! An additional thanks to the reviewers who have already taken our rebuttal into consideration and the area chairs for managing the review of our paper. As we near the end of the discussion period, let us know if there is anything else we can do to improve your evaluation of our work.

---

### Decision · Program_Chairs · 2023-09-21

**Decision:**

Accept (poster)

**Comment:**

In this paper, the authors generalize Plug-and-Play approaches to solving blind inverse problems.  The approach uses two denoising algorithms, one for the image prior and one as a prior for the unknown measurement operator.  The authors extend existing theoretical convergence results to the non-convexity that arises due to blindness, among other novelties.  The provides results show good performance of the propose method relative to baselines in two blind inverse problems.  The strength of this paper is the nontrivial extension of existing PnP techniques, including experimental and theoretical results, to the challenging nonconvex setting of blind inverse problems.  Weaknesses of the original submission involve some missing baselines, but those were addressed during rebuttal.